# Fracture strength analysis of titanium insert-reinforced zirconia abutments according to the axial height of the titanium insert with an internal connection

**Seung-rye Song[1], Kyeong-Mee Park[2], Bock-Young Jung[3]***

**1** Department of Advanced General Dentistry, Dankook University College of Dentistry, Cheonan, Korea,
**2** Department of Advanced General Dentistry, Human Identification Research Institute, Yonsei University College of Dentistry, Seoul, Korea, **3** Department of Advanced General Dentistry, Yonsei University College of Dentistry, Seoul, Korea

* jby1004@yuhs.ac

**Data Availability Statement:** All relevant data are within the manuscript and its Supporting Information files.

## Abstract

This study aimed to analyze fracture strength *in vitro* by varying the axial height of the titanium insert and the labial height of the zirconia abutment in an internal connection implant to identify the titanium insert axial height with optimal mechanical stability. Sixty implants with an internal connection system were used. Two-piece zirconia abutments were used with the titanium inserts. Combinations of different titanium insert axial heights (mm) and zirconia abutment labial heights (mm) constituted five groups: Gr1 (1–3), Gr2 (3–3), Gr3 (3–5), Gr4 (5–3), and Gr5 (5–5). After thermocycling, a fracture load test was performed with a universal testing machine. The initial deformation load and the fracture load were measured and analyzed. The fractured surface and cross-section of the specimens were examined by scanning electron microscopy (SEM). The groups of titanium inserts with axial heights of 3 mm and 5 mm showed significantly greater initial deformation load and fracture load than the group with an axial height of 1 mm ($p < 0.05$), but there was no significant difference between the two groups with axial heights of 3 mm and 5 mm. The labial height of the zirconia abutment had no significant influence on the initial deformation load and fracture load. In some specimens in Gr4 and Gr5, cracking or bending of the titanium insert and abutment screw was observed on SEM. The axial height of the titanium insert should be designed to not be less than 3 mm to increase the fracture strength and promote the long-term stability of implants.

## Introduction

Titanium insert-reinforced two-piece zirconia abutments have been developed and used to complement the mechanical stability of one-piece zirconia abutments [1].

A zirconia abutment with a tooth-like color represents aesthetic advantages in preventing unnatural metal shadows appearing through thin gingival tissue and the exposure of titanium abutments after gingival recession [2, 3]. In addition to the aesthetic advantages, zirconia

**Funding:** This study was supported by a faculty research grant from the Yonsei University College of Dentistry (No 6-2018-0027).

**Competing interests:** The authors have declared that no competing interests exist.

abutments are known to exhibit high fracture resistance and excellent mechanical properties compared to other types of ceramic abutments and allow less bacterial adhesion than both gold and titanium [4, 5]. A number of studies have proven the relevance of the clinical application of zirconia in terms of multiple aspects, such as biocompatibility, mechanical properties and aesthetics [2–4].

Nevertheless, mechanical limitations of one-piece zirconia abutments have also been reported. One thing to consider is the high brittleness of zirconia. Zirconia is often considered a metallic ceramic because of its high strength. However, it is more vulnerable to fracture than conventional titanium abutments when a thin piece of zirconia is used in a prosthesis. Particularly for internally connected types of one-piece zirconia abutments, the thickness of the zirconia abutment is thin at the site of connection inside the fixture. Regarding preloading, the joint stability could be established by applying a preload to obtain the clamping force using the tension between a screw and a component in the titanium abutment; however, zirconia abutments do not allow the plastic deformation of components to determine a safety value for the joint screw and may fracture due to stress concentration [6, 7].

In addition, zirconia is known to undergo low-temperature degradation in a moist environment, which weakens its mechanical properties [8, 9]. Zirconia abutments have fracture strength to the extent that they can withstand physiological occlusal pressure in a certain short-term period but show decreased fracture strength after a 5-year artificial aging process, including changes in the oral temperature and dynamic loading [10, 11].

The marginal gap at the point of contact between a zirconia abutment and titanium implant may be defective due to an error in the abutment manufacturing process or long-term clinical use. A poor fit between the abutment and implant can lead to mechanical complications, such as fracture, loosening of the abutment screw, and deformation of the components, as well as biological complications, such as bone resorption because of stress concentration in the marginal area and peri-implantitis due to bacterial retention [12]. Since prefabricated zirconia abutments are produced by a standardized, precise manufacturing process, the fit with fixtures is excellent. However, precisely fabricating abutments may be difficult, particularly in cases of customized zirconia abutments, owing to errors and sintering shrinkage [13]. Especially for internally connected types, it has been reported that there are few available laboratories with a program for the machining process to cut the abutment according to the internal hexagonal structure unique to the implant fixtures of various companies [14]. According to one study, the vertical gap between the internally connected type of customized zirconia abutment and implant was 5.7–11.8 μm, which was 3 to 7 times greater than that of the titanium abutment [12].

A defect in the fit of the zirconia abutment–implant interface resulting from long-term clinical use may be induced by both a difference in hardness between the zirconia and titanium and the micromobility generated during mastication or deglutition [15]. The hardness of zirconia is approximately 6 to 10 times greater than that of titanium, and micromobility causes titanium implants to wear. As a result, a decrease in fit brings about mechanical and biological complications, namely, loosening of the abutment screw, fracture or deformation of the components, peri-implantitis, and bone resorption [16].

A titanium insert, as a complementary component, was designed and attached to the bottom of the zirconia abutment and allowed the zirconia abutment to be connected to the implant with titanium-to-titanium contact, ensuring force distribution along the internal connection and overcoming mechanical limitations related to inherent problems of zirconia material and processing [1, 3–5, 9, 11, 15].

The mechanical stability, including the flexural strength, fatigue load and fracture resistance, of titanium insert-reinforced zirconia abutments was proven to be significantly higher

than that of one-piece zirconia abutments [3, 5]. The positive effect of the titanium insert was demonstrated by the higher flexural strength of the internal connecting type compared to the external connecting type [11]. However, the use of a titanium insert-reinforced zirconia abutment could lead to the zirconia prosthesis becoming fractured or dislodged.

The factors that influence the mechanical strength of titanium insert-reinforced zirconia abutments are the height, diameter and thickness of the titanium insert and the method of retention. Concerning the efficacy of the retention method, one investigation compared three groups of titanium inserts and zirconia abutments joined with bonded retention, friction fit retention, and ring friction and treated with cyclic loading; they found that the bonded group showed the highest fracture strength, followed by the friction fit group and ring friction group in order [9]. In real clinical situations, cemented retention with composite resin cement between the titanium insert and zirconia abutment has been widely used. Increasing the vertical marginal thickness of the titanium insert to establish structural integrity does not correspond to the purpose of the fabrication of the zirconia abutment to avoid a metallic appearance. However, the axial height of the titanium insert could determine the surface area bonded to the zirconia abutment and varies depending on complicated clinical conditions, such as the angle of implant placement, gingival thickness, and crown height space. However, there are few studies in the literature that can provide guidelines for the fabrication of titanium inserts.

This *in vitro* study was designed to measure the fracture strength and observe the fracture patterns of titanium insert-reinforced zirconia abutments fabricated by varying the axial height of the titanium insert and the labial height of the zirconia abutment to suggest an ideal titanium insert axial height in terms of the integrity of the abutment–implant complex.

## Material and methods

Sixty screw-shaped implants 4.0 mm in diameter and 10.0 mm in length (Implantium, Dentium, Seoul, Korea) were used, and a total of sixty titanium-reinforced zirconia abutments were divided into five groups based on the axial height of the titanium insert and the labial height of the zirconia abutment in this study. The implants tested in this study were composed of SLA surface-treated pure titanium and were consistent with those used clinically.

### Test specimen design

The bottom of the titanium insert was 4.0 mm in diameter, and the axial height from the bottom to the top (Fig 1A) varied from 1 mm to 3 mm and 5 mm; these inserts were fabricated using a computer numerical control (CNC) machining center (L20, Cincom, Japan). All zirconia abutments were fabricated to be 13.0 mm in length from the titanium insert–abutment junction to the top of the abutment assuming use in a maxillary anterior prosthesis by CAD/CAM with a labial height of 3 mm or 5 mm to evaluate the effect of the labial height of the zirconia abutment (Fig 1B). Meanwhile, the height of the palatal side was set to be equal in all specimens (Fig 1).

A total of 60 specimens were divided into 5 groups, with 12 specimens in each group, based on the axial height of the titanium insert and the labial height of the zirconia abutment: Gr1 (1–3), Gr2 (3–3), Gr3 (3–5), Gr4 (5–3), and Gr5 (5–5).

### Preparation of specimens

To assemble the titanium insert and zirconia abutment, the inner surface of the zirconia abutment was treated with silica-modified aluminum oxide sandblasting (Rocatec; 3M ESPE) to enhance the bonding strength with the titanium insert, and then a ceramic primer (Monobond

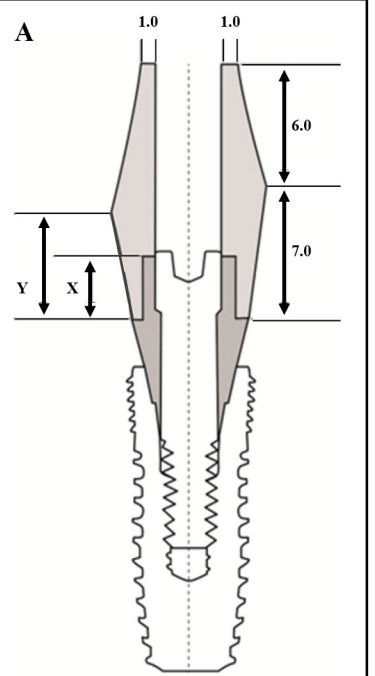
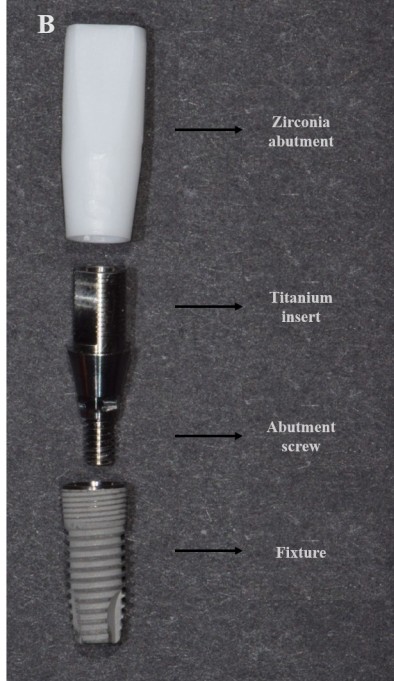

**Fig 1. Titanium insert and zirconia abutment.** (A) Schematic diagram. (B) Specimen design. X, Axial height of the titanium insert. Y, Labial height of the zirconia abutment (Unit: mm).

Plus; Ivoclar Vivadent, Schaan, Liechtenstein) was applied. The zirconia abutment was bonded with the titanium insert using dual-cure self-adhesive resin cement (RelyX Unicem, 3M ESPE, St. Paul, MN, USA) according to the manufacturer's instructions. Remaining adhesive was removed under a microscope at 10X magnification.

The abutment was connected to the implant through a titanium abutment screw using a torque wrench (Dentium, Seoul, Korea) with 30 Ncm of tightening torque, as recommended by the manufacturer. Implant–abutment joint stability was ensured by retightening the screw to establish sufficient preloading [6, 17].

Then, the abutment was stored in distilled water at 37˚C for 24 hours to eliminate the effect of volume change due to adhesive hydration. Thermocycling (Thermal Cyclic Tester, Thermocycling: R&B, Inc.) was performed at 5–55˚C in a water bath at intervals of 15 seconds and 10 seconds to simulate five years of exposure to moist oral conditions [18].

## Load test

To investigate the fracture strength in this test set up, the abutment–implant specimen was fixed to the universal testing machine (Model 3366; Instron®, Norwood, MA, USA) using a metal holder that was positioned with a vertical distance of 3 mm downward from the platform of the implant to simulate 3 mm of crestal bone resorption [19]. The load was applied on the palatal side of the specimen with a 30˚ oblique direction to simulate the off-axis loading of anterior teeth during functions such as mastication or chewing at a crosshead speed of 0.5–1.0 mm/min. The test was carried out until the specimen fractured or significantly deformed (Fig 2).

**Initial deformation load.** Although the load–deformation curve shows progressive proportionality, the sudden reduction of the load in some segments indicates cracking, bending

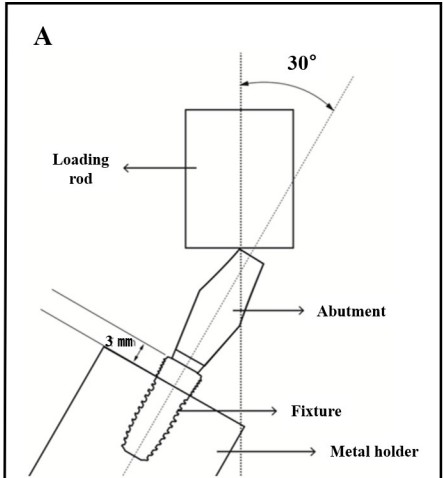
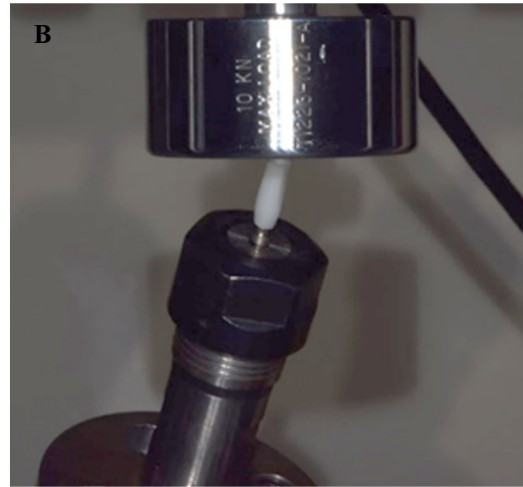

**Fig 2. Fracture load test using universal testing machine.** (A) Angle of force. (B) Clinical photograph of the fracture load test.

or loosening of the component, which means a potential failure in the function of the prosthesis [20–22]. In this study, the load applied at the initial point where the load markedly dropped in the load–deformation curve was measured as the initial deformation load (Fig 3A).

**Fracture load.** The fracture load was automatically recorded at the point where the fracture occurred through Instron Series IX Software (Fig 3B).

## Observation of fracture patterns

**Observation of fractured surface by scanning electron microscopy (SEM).** The fractured surface of three specimens from each group was observed under a scanning electron microscope (Hitachi S-3000N, Hitachi3000N, Tokyo, Japan). The superior aspect of the implant (20X magnification) was observed to determine the damage in the inner hexagonal surface of the titanium insert. The lateral aspect of the connecting part of the fixture and abutment (18X and 30X magnification) was observed to verify deformation in the titanium insert, abutment, and fixture. The surface of fractured zirconia (40X magnification) was observed to examine the fracture patterns of the zirconia abutment.

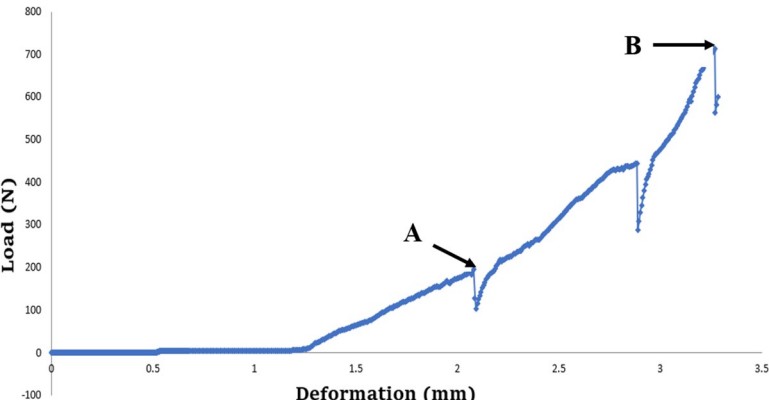

**Fig 3. Load–deformation curve.** (A) Initial deformation load. (B) Fracture load. N: Newton.

**Table 1. Initial deformation load in each group according to the axial height of the titanium insert.**

| Group | Gr1 | Gr2 | Gr3 | Gr4 | Gr5 |
|---|---|---|---|---|---|
| n | 8 | 12 | 9 | 11 | 11 |
| Axial height | 1 mm | 3 mm | 3 mm | 5 mm | 5 mm |
| N | 89.19 ± 52.50 | 190.46 ± 91.20 | 179.86 ± 35.08 | 180.63 ± 72.76 | 177.88 ± 39.65 |
| | 89.19 ± 52.50 | 185.92 ± 71.38 | | 179.26 ± 57.20 | |

n: number of specimens, N: Newton (mean ± standard deviation).

**Observation of cross-section by microscope.** Blocks were fabricated by burying two specimens from each group in acrylic resin (Technovit 7200 VLC, Kulzer, Wehrheim, Germany) to determine the deformation pattern of the abutment–implant interface after the load test. Specimens were prepared by cutting along the longitudinal axis of the implant and were observed under a microscope (OLYMPUS BX43, UIS2 optical system, Olympus Corporation, Tokyo, Japan); 12.5X, 40X, 100X.

## Statistical analysis

The initial deformation load and fracture load were processed statistically using SPSS (SPSS Version 26.0, IBM SPSS, Inc., Chicago, IL, USA). The power of the experiment was sufficiently high to show that all differences were statistically significant. A power analysis was therefore not required. The nonparametric Kruskal-Wallis test and Mann-Whitney $U$ test were used for data analysis. Bonferroni correction was used to adjust the confidence interval. The level of statistical significance was set at less than 1.7% ($P$-value <0.017).

## Results

A total of nine zirconia abutment–titanium insert specimens were excluded from the analysis due to not only abutment dislodgement from the titanium insert (one in Gr3 and one in Gr5 during screwing into the implants and two in Gr1 during the thermocycling procedure) but also zirconia fracture at an early stage of the load test (two in Gr1, two in Gr3, and one in Gr4). Therefore, the test results of fifty-one specimens were subjected to analysis: Gr1 (n = 8), Gr2 (n = 12), Gr3 (n = 9), Gr4 (n = 11), and Gr5 (n = 11).

### Fracture load test

**Initial deformation load.** The mean initial deformation load in each group was 89.19 N, 190.46 N, 179.86 N, 180.63 N, and 177.88 N in Gr1, Gr2, Gr3, Gr4 and Gr5, respectively (Table 1). According to the axial height of the titanium insert, the mean initial deformation load in the 1 mm group (Gr1) was 89.19 N, that in the 3 mm groups (Gr2 and Gr3) was 185.92 N, and that in the 5 mm groups (Gr4 and Gr5) was 179.26. The 1 mm group showed a significantly lower initial deformation load than the 3 mm and 5 mm groups (Table 2). There was no

**Table 2. Multiple pairwise comparisons of the initial deformation load by the axial height of the titanium insert.**

| Multiple comparisons | | Mann-Whitney $U$ | $P$-value |
|---|---|---|---|
| Gr1 –Gr2, 3 | 1 mm– 3 mm | 21.0 | 0.002 * |
| G1 –G4, 5 | 1 mm– 5 mm | 20.0 | 0.001 * |
| G2, 3 –G4, 5 | 3 mm– 5 mm | 209.0 | 0.593 |

* $P$-value <0.017 (adjusted by Bonferroni's method).

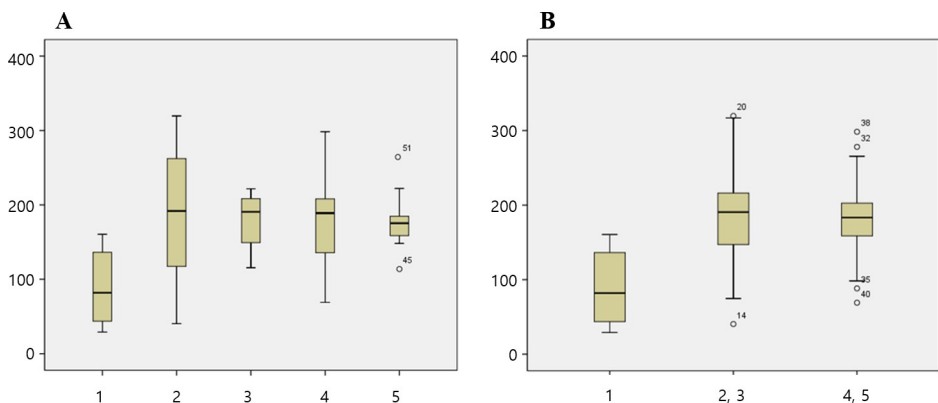

**Fig 4. Box plot of the initial deformation load.** (A) Comparison of five groups. (B) Comparison according to the axial height. X axis of the graph: group, Y axis of the graph: Newton.

significant difference between the 3 mm and 5 mm groups (Fig 4). There was no significant difference in the initial deformation load according to the labial height of the zirconia abutment (Fig 5).

**Fracture load.** The mean fracture load in each group was 361.7 N, 509.07 N, 558.5 N, 814.88 N, and 608.54 N in Gr1, Gr2, Gr3, Gr4 and Gr5, respectively (Table 3). According to the axial height of the titanium insert, the mean fracture load in the 1 mm group (Gr1) was 361.7 N, that in the 3 mm groups (Gr2 and Gr3) was 530.26 N, and that in the 5 mm groups (Gr4 and Gr5) was 711.73 N. The titanium insert groups with axial heights of 3 mm and 5 mm showed significantly higher fracture loads than the 1 mm group, and there was no significant difference between the 3 mm and 5 mm groups (Table 4, Fig 6). There was no significant difference in the fracture load according to the labial height of the zirconia abutment (Fig 7).

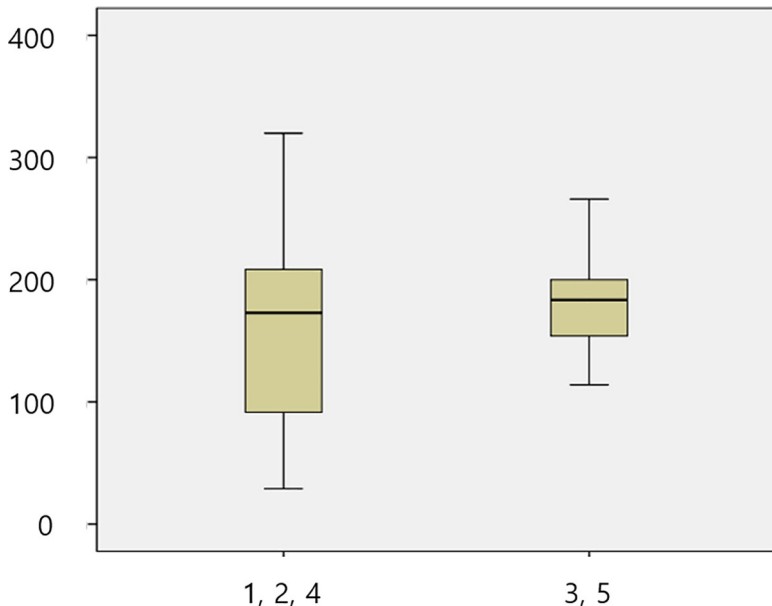

**Fig 5. Box plot of the initial deformation load by the labial height.** Comparison according to the labial height of the zirconia abutment. X axis of the graph: group, Y axis of the graph: Newton.

**Table 3. Fracture load in each group according to the axial height of the titanium insert.**

| Group | Gr1 | Gr2 | Gr3 | Gr4 | Gr5 |
|---|---|---|---|---|---|
| n | 8 | 12 | 9 | 11 | 11 |
| Axial height | 1 mm | 3 mm | 3 mm | 5 mm | 5 mm |
| N | 361.75 ± 2.41 | 509.07 ± 216.22 | 558.50 ± 213.50 | 814.88 ± 412.73 | 608.54 ± 251.20 |
| | 361.7 ± 52.41 | 530.26 ± 211.12 | | 711.7 ± 349.77 | |

n: number of specimens, N: Newton (mean ± standard deviation).

## Patterns of fracture

The zirconia abutment completely dislodged from the titanium insert or partially fractured (Fig 8). In Gr1, the zirconia abutment dislodged from the titanium insert in all specimens. There were completely dislodged or partially fractured specimens in Gr2, Gr3, Gr4, and Gr5 (Table 5). All crack lines involved the palatal cervical portion and the abutment–titanium insert interface. In all partially fractured specimens, fractures along the longitudinal direction appeared from the upper surface of the titanium insert. The number of specimens in which the zirconia abutment completely dislodged decreased as the axial height of the titanium insert increased (Table 5).

**Characteristics of the fractured surface.** The fractured surface of three specimens from each group was observed under a SEM. A superior and lateral view of the implant and the fractured surface of the zirconia abutment were examined (Fig 9). In the superior view of the titanium insert, fine wear and scratches on the inner hexagonal screw surface were observed. In the lateral view, wear and scratches were observed on the surface of both the titanium insert and the zirconia abutment. Cracks were observed on the surface of the titanium insert in one specimen in Gr4, and the hackle (region where the tensile stress tilted from the crack surface) was shown as the cracks in zirconia progressed [23]. Fragments of zirconia and residual adhesive were noticed on the surface of the titanium insert, and the amount and pattern of residual adhesive varied by specimen. Wear, scratches, and residual adhesive were observed on the fractured surface of the zirconia abutment.

**Characteristics of cross-section.** Two specimens from each group were cut along the long axis of the specimen, and the sectioned surface was observed under a microscope (Fig 10). Meanwhile, marked deformation was not identified in the titanium inserts and abutment screws of samples from Gr1 and Gr2 (Fig 10). Both gap and bending deformation of the titanium inserts and abutment screws were also exacerbated in Gr3, Gr4 and Gr5. Higher magnification (40X and 100X magnification) SEM images of samples from Gr5 demonstrated an oblique crack line in the abutment screw head in contact with the titanium insert even though the titanium insert was not clearly deformed (Fig 10).

## Discussion

This study shows that a zirconia abutment reinforced by a titanium insert with an axial height of at least 3 mm could resist the biomechanical stress exerted during oral functions, such as

**Table 4. Multiple pairwise comparisons of the fracture load by the axial height of the titanium insert.**

| Multiple comparisons | | Mann-Whitney $U$ | $P$-value |
|---|---|---|---|
| Gr1 –Gr2, 3 | 1 mm– 3 mm | 28.0 | 0.006 * |
| G1 –G4, 5 | 1 mm– 5 mm | 14.0 | 0.001 * |
| G2, 3 –G4, 5 | 3 mm– 5 mm | 143.0 | 0.033 |

* $P$-value <0.017 (adjusted by Bonferroni's method).

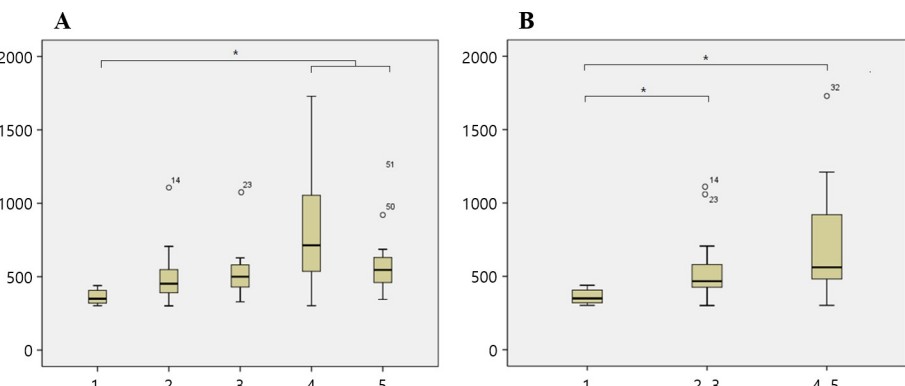

**Fig 6. Box plot of the fracture load test.** (A) Comparison of five groups. (B) Comparison according to the axial height. X axis of the graph: group, Y axis of the graph: Newton.

mastication and chewing. Physiological masticatory force developed during mastication and deglutition is known to be 90–370 N for anterior teeth and, 200~900 N for posterior teeth, while 300 N has been suggested as an appropriate load for a single premolar and 700 N as a maximum load for the second molar region, with a monotonic increase in the load along the dental arch; however, there are still conflicting findings [24–28]. Due to the lack of periodontal proprioceptors, a higher functional load is expected on implant–supported fixed dental prostheses than natural teeth [14].

Considering the low mean initial deformation load (89.19 N) and fracture load (361.7 N) of zirconia abutments with a titanium insert with an axial height of 1 mm in this study, a titanium insert with an axial height of 1 mm seems inappropriate for clinical application due to the low strength regardless of the location of the prosthesis.

Since the mean initial deformation load in the 3 mm and 5 mm titanium insert groups was 185.92 N and 179.26 N and the fracture load in the 3 mm and 5 mm titanium insert groups

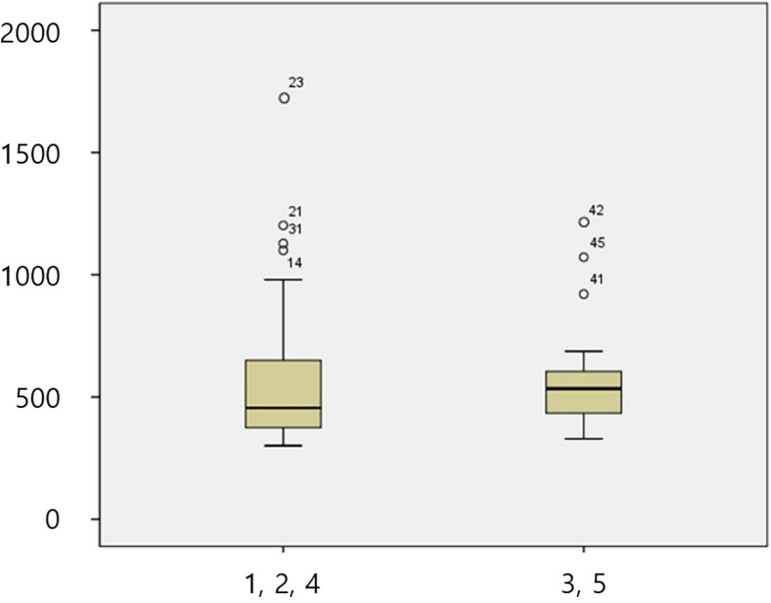

**Fig 7. Box plot of the fracture load test by the labial height.** Comparison according to the labial height of the zirconia abutment. X axis of the graph: group, Y axis of the graph: Newton.

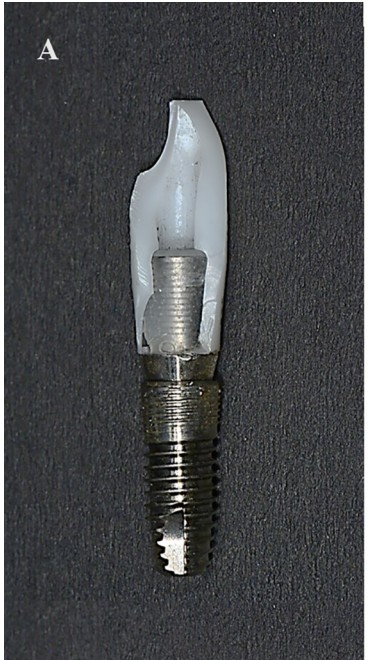 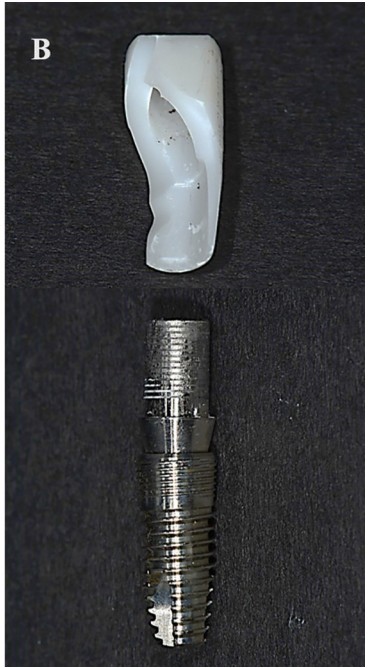

**Fig 8. Fracture pattern of the zirconia abutment after the fracture load test.** (A) Partial fracture. (B) Exfoliation.

was 530.26 N and 711.70 N, respectively, a titanium insert with an axial height of 3 mm or 5 mm was proven to be suitable for anterior teeth replacement. A study of the effect of various cements on the retentiveness of zirconia abutments with respect to the titanium insert axial height reported that there was a significant increase regardless of the cement used and that the axial height of titanium inserts should be at least 3 mm to maintain zirconia abutment retention [29].

However, in the case of posterior restorations, it is necessary to design prostheses to reduce stress by reducing the size of the occlusal table or lowering the cusp inclination of crowns. One previous study concluded that a zirconia abutment with a titanium insert appeared to be a suitable treatment option in the anterior and premolar regions after an observation period of 6 years [30]. However, the masticatory force may vary depending on the patient's age, sex, diet, opposing dentition, and remaining teeth. Parafunctions, such as bruxism, clenching, and tongue thrusting, impose greater stress on the teeth. According to Gibbs et al., for adult men with parafunctions, the load on maxillary anterior teeth has been reported to reach up to 720 N [27]. Thus, the use of zirconia abutments with titanium inserts should be restricted in patients with parafunctions or high masticatory forces based on the results of this study.

**Table 5. Fracture pattern distributions in each group.**

|  |  | Gr1 | Gr2 | Gr3 | Gr4 | Gr5 |
|---|---|---|---|---|---|---|
| Exfoliation | n | 8 | 4 | 5 | 2 | 2 |
|  | % | 100 | 33.3 | 55.6 | 18.2 | 18.2 |
| Partial fracture | n | 0 | 8 | 4 | 9 | 9 |
|  | % | 0 | 66.7 | 44.4 | 81.8 | 81.8 |

n: number of specimens, %: percentage.

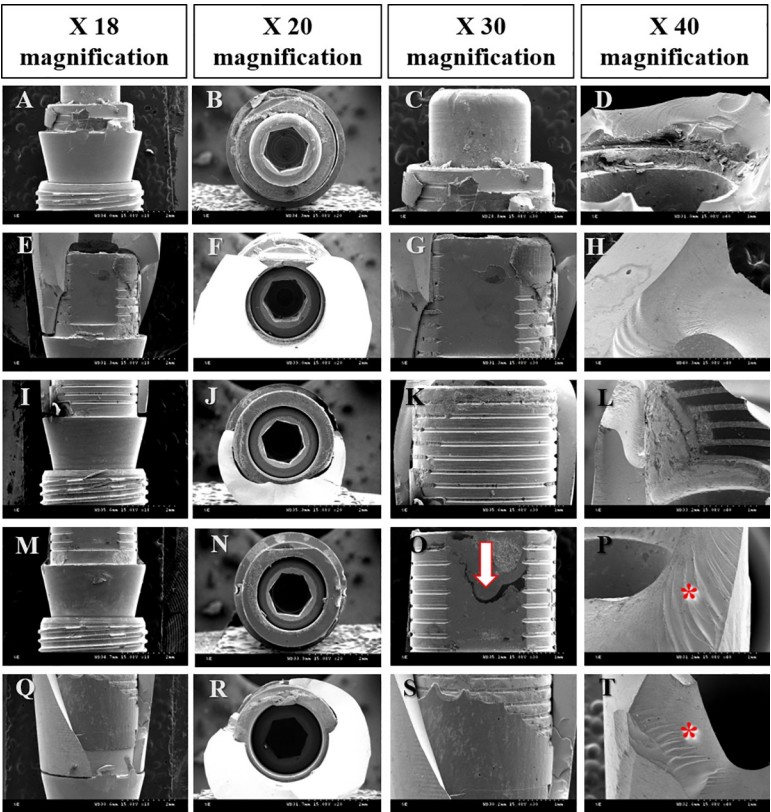

**Fig 9. SEM images of the implant surface after the fracture load test.** (A-D) Gr1. (E-H) Gr2. (I-L) Gr3. (M-P) Gr4. (Q-T) Gr5. Arrow, crack in the titanium insert. Asterisk, fractured surface of the zirconia abutment.

The vertical crown height or interocclusal space is also an important factor to consider when designing titanium insert reinforced zirconia abutments. The interocclusal space refers to the distance between the ridge crest and the occlusal plane comprising the depth of the soft tissue above the fixture, the height of the abutment, and the thickness of the implant prosthesis [17]. Based on the results of this study showing that the axial height of the titanium insert should be 3 mm or more to ensure structural integrity, a limited interocclusal space can be a contraindication for the use of titanium insert-reinforced zirconia prosthesis unless the margin of the zirconia prosthesis is located subgingivally.

The initial deformation load was measured in addition to the fracture load. In the early stage of the load–deformation curve, a small amount of fluctuation could be seen, the first point of which was the initial deformation load. This point could be considered to indicate the potential loss of functional stability because it signifies a change in the original structural integrity, such as initial cracking or bending, began to occur due to external compressive loading. Even if the fracture load is high, a restoration with a low initial deformation load is prone to fracture under cyclic fatigue loading. The conventional load–deformation curve shows a sharp drop in load at the end of the curve, indicating the fracture or plastic deformation of components [31], but in this study, there were comparatively frequent fluctuations in the load on the overall curve because the structure of the titanium insert-reinforced zirconia abutment is completely different from that of other abutments made of one material, such as a titanium or zirconia abutment. In this study, the initial deformation load measured in all groups was less than 200 N, and the titanium insert-reinforced zirconia abutment could be inferred to have

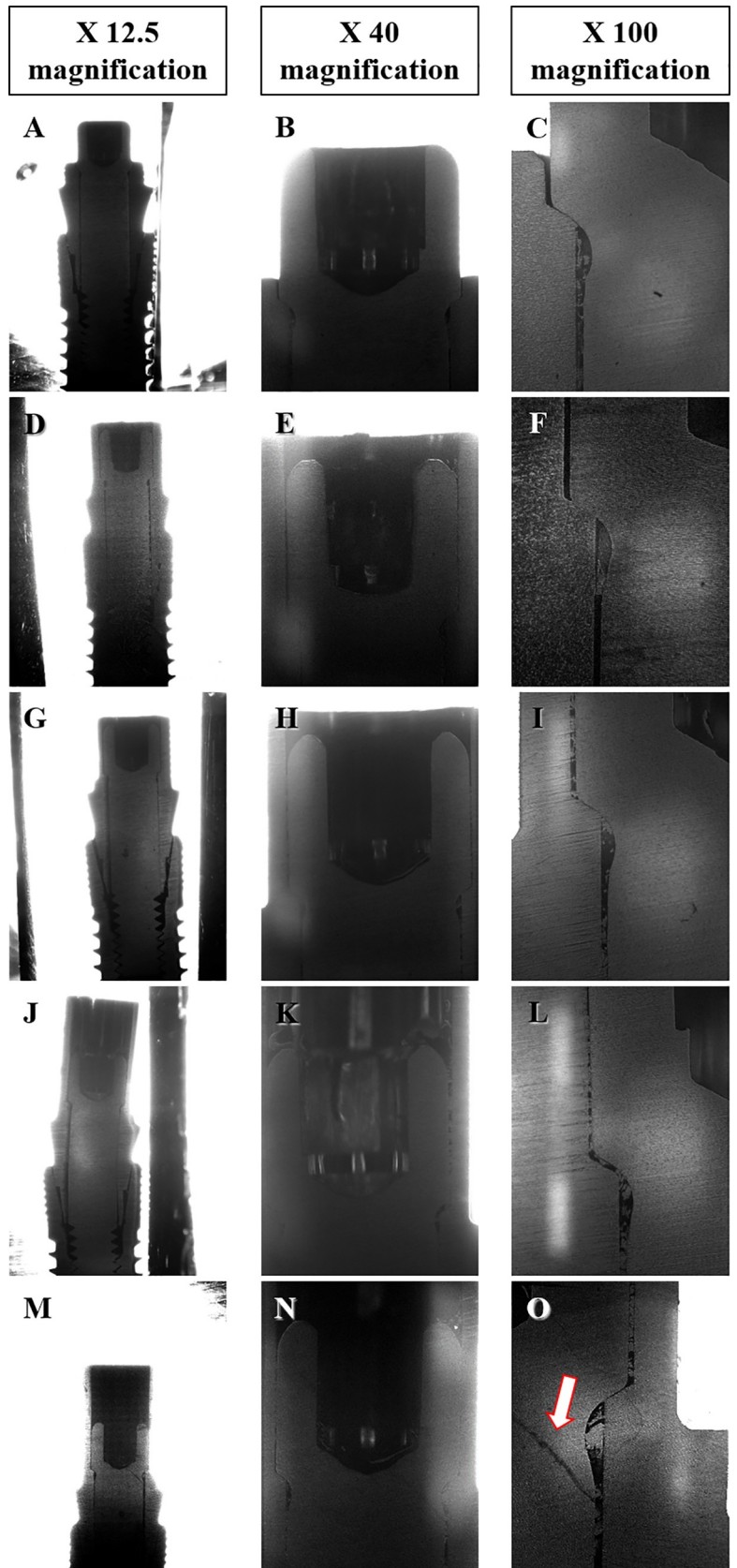

**Fig 10. SEM images of a cross-section of the implant after the fracture load test.** (A-C) Gr1. (D-F) Gr2. (G-I) Gr3. (J-L) Gr4. (M-O) Gr5. Arrow, crack in an abutment screw.

potential failure factors considering the practical masticatory force [24–28]. Therefore, it is necessary to avoid repetitive non-protective forces on the implant, and it is recommended to avoid the use of titanium insert-reinforced zirconia abutments in any regions under a high occlusal load.

The values of the load test did not significantly change with changes in the labial height of the zirconia abutment in this study. It can be concluded that the labial height of zirconia is unrelated to the fracture load and initial deformation load of titanium insert-reinforced zirconia abutments.

Previously published studies have investigated the effect of the zirconia abutment thickness on the fracture strength and fatigue load, which was not considered in this study, and reported that the fracture strength was highest in the 0.5 mm group; additionally, there was a significant difference between this group and other groups with preparation depths of 0.7 and 0.9 mm, suggesting that a zirconia abutment preparation depth of 0.5 mm is advisable for titanium insert-reinforced zirconia abutments [15, 32]. Although the fracture strength of zirconia abutments with a preparation depth of 0.5 mm, 0.7 mm and 0.9 mm showed values above physiological mastication loads and other research has demonstrated no significant differences among groups in the fracture strength of specimens with a preparation depth of less than 0.9 mm [33], a preparation depth above 0.7 mm was not recommended.

It was inferred that a higher torque and stress is expected on the area of contact between the thin zirconia wall and the axial titanium wall than on the interface between the screw head and abutment or the abutment and implant; therefore, the volume of the labial side of the zirconia abutment could affect the fracture strength comparison, based on reports in the literature indicating that the labial side of an anterior zirconia abutment is 1.7 times stronger under compression than the palatal surface under tensile stress [34]. An initial null hypothesis in this study was that the labial height of the zirconia abutment affects the fracture strength, but this hypothesis was rejected. After all, the thickness, rather than the height, of the zirconia abutment influences the fracture resistance.

In Gr1, with the lowest titanium insert axial height, all zirconia abutments completely detached from the titanium inserts during the fracture load test without deformation of the titanium insert, abutment screw, or fixture on SEM. It was assumed that this result was caused by low retention force due to the short axial height on the basis of an investigation of changes in the retention force with varying abutment height [35]. With increasing titanium insert axial height, more specimens showed partial fracture in the zirconia abutment. Fracture of the zirconia abutment without deformation of the titanium insert means that the stress reached the fracture strength of the zirconia abutment before it concentrated at the insert–implant interface [3]. From the results of this study, deformation of implant components, such as cracking or bending of the titanium insert and abutment screw, was observed in specimens in Gr4 and Gr5, with a titanium insert axial height of 5 mm, which showed a high fracture load.

The fractured zirconia abutments showed a vertical fracture pattern in the longitudinal direction of the implant, and in all fractured specimens, the cervical portion of the abutment was involved. It could be inferred that cracks started from the cervical portion of the abutment due to tensile stress transformed from compressive force on the top of the titanium insert and propagated along the abutment. At this point, the axial height of the titanium insert is presumed to play an important role in the fracture strength of the abutment.

The present study was not designed to determine whether fatigue loading affects the fracture strength and integrity of the abutment–implant complex. Considering the initial

deformation load, under cyclic loading, specimens with a low initial deformation load are more prone to be vulnerable to deformation of the abutment screw or the abutment itself. As factors related to the retentiveness of the abutment and the initial deformation load, the height, surface area, slope, surface treatment, and adhesive could be considered [29, 36], but little information is available. Further investigation into the retentiveness of titanium inserts and zirconia abutments under cyclic fatigue loading to simulate actual dynamic functions, such as chewing and mastication, will increase the success rate of implant-supported fixed prostheses with titanium insert-reinforced zirconia abutments for aesthetic purposes.

Thermocycling treatment in this study was performed to evaluate clinical stability by simulating five years of exposure to moist oral conditions [9, 37]. Many studies have reported that zirconia oxide (3Y–TZP) is very susceptible to moisture and is weakened through low-temperature degradation, and an increased fracture rate was observed under humidity exposure [32]. A total of nine specimens were excluded owing to early-stage failure in this experiment, and two factors, i.e., low-temperature degradation of zirconia oxide and low bonding strength between the zirconia abutment and titanium insert, are thought to be the main causes.

The uneven and small numbers of specimens per group (Gr 1 (8), Gr 2 (12), Gr 3 (9), Gr 4 (11), and Gr 5 (11)) is considered one of the limitations of this study. To avoid bias in the data analysis, nine failed specimens in the preparation stages for the screwing, thermocycling, and loading test were excluded. In addition, non-anatomical, internal connection type abutments were used in this study. Further experimental studies using anatomically designed zirconia prostheses and large-size specimens are required to achieve more clinically meaningful results. An experimental study is needed to investigate the fatigue load and labial thickness of titanium insert-reinforced zirconia abutments to maintain the integrity of the abutment complex under dynamic functional loading using a greater number of specimens and a load applicator to mimic the conditions of occlusion between anterior teeth, followed by supporting clinical research.

## Conclusion

Within the limitations of this *in vitro* study, the conclusion regarding clinical use is that the axial height of the titanium insert should be designed to not be less than 3 mm to increase the fracture strength and promote the long-term stability of implant-supported fixed prostheses with titanium insert-reinforced zirconia abutments. In addition, considering the initial deformation load, the occlusion scheme and prosthetic structure should be designed to reduce stress and distribute the functional load on implant-supported prostheses for anterior teeth.

## Supporting information

**S1 Data. The set of data in this study.**
(XLSX)

## Author Contributions

**Conceptualization:** Bock-Young Jung.

**Data curation:** Seung-rye Song, Kyeong-Mee Park.

**Funding acquisition:** Bock-Young Jung.

**Investigation:** Bock-Young Jung.

**Project administration:** Bock-Young Jung.

**Validation:** Kyeong-Mee Park.

**Visualization:** Seung-rye Song, Kyeong-Mee Park.

**Writing – original draft:** Seung-rye Song.

**Writing – review & editing:** Kyeong-Mee Park, Bock-Young Jung.

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
