## [Decision Letter · Decision Letter 0]

22 Feb 2021

PONE-D-21-03699

Fracture strength analysis of titanium insert reinforced zirconia abutment by axial height of titanium insert with internal connection

PLOS ONE

Dear Dr. Jung,

Thank you for submitting your manuscript to PLOS ONE. After careful consideration, we feel that it has merit but does not fully meet PLOS ONE’s publication criteria as it currently stands. Therefore, we invite you to submit a revised version of the manuscript that addresses the points raised during the review process.

We look forward to receiving your revised manuscript.

Kind regards,

Antonio Riveiro Rodríguez, PhD

Academic Editor

PLOS ONE

Reviewers' comments:

Reviewer's Responses to Questions

**Comments to the Author**

1. Is the manuscript technically sound, and do the data support the conclusions?

Reviewer #1: Yes

Reviewer #2: Yes

2. Has the statistical analysis been performed appropriately and rigorously? 

Reviewer #1: Yes

Reviewer #2: Yes

3. Have the authors made all data underlying the findings in their manuscript fully available?

Reviewer #1: Yes

Reviewer #2: Yes

4. Is the manuscript presented in an intelligible fashion and written in standard English?

Reviewer #1: No

Reviewer #2: Yes

5. Review Comments to the Author

Reviewer #1: There is no specification of implant type regarding sterile and clinical usage type of implants, while those implants on the figures attached have smooth surface and they should not be used for clinical purposes, the surface should be rough and specific related to the manufacturer! This in vitro study should be performed on the original, sterile packed dental implants prepared for clinical use.

English should be carefully checked by the native speaker and the manuscript need serious proof editing before final consideration for publication.

Reviewer #2: I have reviewed the manuscript Fracture strength analysis of titanium insert reinforced zirconia abutment by axial height of titanium insert with internal connection” submitted to “PLOS ONE” for publication. In this study, authors have analyzed the fracture strength in vitro by varying the axial height of the titanium insert and the labial height of the zirconia abutment in an internal connection implant to identify the titanium insert axial height with optimal mechanical stability. The manuscript fits well within the scope of the journal; it needs some major improvements; there are a few suggestions that authors may consider improving it further:

The use of English language is reasonable, however, there are a number of minor punctuation and grammatical errors; that should be corrected and rephrased using academic English for a better flow of text for reader.

Abstract: covered all key information

Introduction; is very detailed and covering the background information and the rationale of the study effectively. Some of the statements have missing citations; such as for this statement

A number of studies have proven the relevance…..: author may cite the following references

Customized therapeutic surface coatings for dental implants. Coatings. 2020 Jun;10(6):568.

Dental Implants: Materials, Coatings, Surface Modifications and Interfaces with Oral Tissues. Woodhead Publishing; 2020 Jul 23.

https://doi.org/10.1016/B978-0-12-819586-4.00002-0

Similarly, the text in the second paragraph of the introduction require citations.

A titanium insert, as a complementary component….. require citations as given above can be used.

Methods: why not used same number of samples after exclusion (n=?) for each group

Gr1 (n=8), Gr2 (n=12), Gr3 (n=9), Gr4 (n=11), Gr5 (n=11).?

Results and data are well explained and beautifully presented.

In discussion: please include discussion in terms of reduced vertical height; or reduced inter occlusal space (deep bite) case. Is there any relation?

What are the limitations of the study?

6. PLOS authors have the option to publish the peer review history of their article (what does this mean?). If published, this will include your full peer review and any attached files.

Reviewer #1: No

Reviewer #2: No

---

## [Author Response · Author response to Decision Letter 0]

3 Mar 2021

1. Title:

REVISION: PONE-D-21-03699

Fracture strength analysis of titanium insert reinforced zirconia abutment according to the axial height of titanium insert with internal connection

2. Corresponding author’s name, mailing address, phone number and fax number:

Prof. Bock -Young Jung

E-mail address: jby1004@yuhs.ac

Phone: (+82)2-2228-8980, Fax: (+82)2-2227-8906

3. Names of authors:

Seung-rye Song, Kyeong-Mee Park, Bock-Young Jung 

This response letter accompanies our online resubmission of the manuscript titled “ Fracture strength analysis of titanium insert reinforced zirconia abutment according to the axial height of titanium insert with internal connection ” for the publication in “ PLOS ONE ”.

We really thank the reviewer’s valuable comments which made our manuscript more sound and reasonable scientifically. We are re-submitting after revisions as requested by the reviewers. We tried to address all the comments carefully in the revised manuscript. We hope that our revised manuscript meets the high standards of your journal.

We hope that this paper would be published in ‘PLOS ONE’. We thank you in advance for your time and attention.

Sincerely yours,

27th February, 2021

Bock Young, Jung

 

Response letter

Manuscript ID: PONE-D-21-03699 

MS-Type: Original Article

Title: " Fracture strength analysis of titanium insert reinforced zirconia abutment according to the axial height of titanium insert with internal connection." 

Correspondence Author: Bock- Young, Jung

Reviewer #1

1. There is no specification of implant type regarding sterile and clinical usage type of implants, while those implants on the figures attached have smooth surface and they should not be used for clinical purposes, the surface should be rough and specific related to the manufacturer! This in vitro study should be performed on the original, sterile packed dental implants prepared for clinical use.

Response: We respect and appreciate the comment of the reviewer. 

We have modified ‘material and methods’ section and specified the information related to the implants tested.

Text Change: See “Material and methods”.

“The implants tested in this study were composed of SLA surface-treated pure titanium and were consistent with those used clinically.”

2. English should be carefully checked by the native speaker and the manuscript need serious proof editing before final consideration for publication 

Response: 

Thank you for the comment. We carefully tried to correct any errors in spelling, grammar, and terminology of this manuscript.

Text Change: The corrected words and phrases were highlighted. 

Reviewer #2

I have reviewed the manuscript Fracture strength analysis of titanium insert reinforced zirconia abutment by axial height of titanium insert with internal connection” submitted to “PLOS ONE” for publication. In this study, authors have analyzed the fracture strength in vitro by varying the axial height of the titanium insert and the labial height of the zirconia abutment in an internal connection implant to identify the titanium insert axial height with optimal mechanical stability. The manuscript fits well within the scope of the journal; it needs some major improvements; there are a few suggestions that authors may consider improving it further:

1. The use of English language is reasonable, however, there are a number of minor punctuation and grammatical errors; that should be corrected and rephrased using academic English for a better flow of text for reader. 

Response:

We appreciated this comments and this manuscript was reedited by one of native experts whose specialty was dentistry according the advice of the reviewer. 

Text Change: See the marked-up manuscript. 

2. Introduction; is very detailed and covering the background information and the rationale of the study effectively. Some of the statements have missing citations; such as for this statement. A number of studies have proven the relevance…..: author may cite the following references. Similarly, the text in the second paragraph of the introduction require citations.

A titanium insert, as a complementary component….. require citations as given above can be used. 

Response:

We appreciate this important comment of the reviewer. We have specified the references we cited.

Text Change: See “Introduction” section.

3. Methods: why not used same number of samples after exclusion (n=?) for each group

Gr1 (n=8), Gr2 (n=12), Gr3 (n=9), Gr4 (n=11), Gr5 (n=11).?

Response:

We appreciated this valuable comment of the reviewer. 

This study was originally prepared by dividing 60 specimens into 5 groups, 12 pieces for each group. However, we excluded nine specimens. Four specimens during the preparation for screwing and thermocycling procedures and 5 specimens during the preparation for loading were failed, which could affect the results of the experiment. 

Text Change: We have modified the end section of discussion and added statement on the reason for the uneven distribution of specimens.

4. In discussion: please include discussion in terms of reduced vertical height; or reduced inter occlusal space (deep bite) case. Is there any relation?

Response

Thank you for the valuable comment. According to the advice of reviewer, we have introduced and added more description regarding the clinical implications that the reduced interocclusal distance has.

Text Change: See “Discussion” in page.

“Vertical crown height or interocclusal space is also an important factor to consider when designing titanium insert reinforced zirconia abutments. The inter occlusal space means the distance between the ridge crest and the occlusal plane comprising the depth of the soft tissue above the fixture, the height of the abutment, and the thickness of the implant prosthesis [17]. Based on the results of this study that the axial height of the titanium insert should be 3mm or more to ensure structural integrity, the limited interocclusal space can be a contra-indication to titanium insert-reinforced zirconia prosthesis unless the margin of zirconia prosthesis is located subgingivally.” 

5. What are the limitations of the study?

Response: 

Thank you for the valid comment of reviewer. We have modified “Discussion” adding several limitations of this study and presented the next research for the further investigation.

Text Change: See “ Discussion” in page.

“ The uneven and small numbers of specimens per group (Gr 1 (8), Gr 2 (12), Gr 3 (9), Gr 4 (11), and Gr 5 (11)) is considered one of the limitations of this study. To avoid bias in the data analysis, nine failed specimens in the preparation stages for the screwing, thermocycling, and loading test were excluded. In addition, non-anatomical, internal connection type abutments were used in this study. Further experimental studies using anatomically designed zirconia prostheses and large-size specimens are required to achieve more clinically meaningful results. An experimental study is needed to investigate the fatigue load and labial thickness of titanium insert-reinforced zirconia abutments to maintain the integrity of the abutment complex under dynamic functional loading using a greater number of specimens and a load applicator to mimic the conditions of occlusion between anterior teeth, followed by supporting clinical research.“

Thank you very much again.

We really thank the reviewer’s comments which really solidify our manuscript more reasonably. 

Sincerely,

Bock Young Jung

(Corresponding author)

---

## [Decision Letter · Decision Letter 1]

15 Mar 2021

Fracture strength analysis of titanium insert-reinforced zirconia abutments according to the axial height of titanium insert with an internal connection

PONE-D-21-03699R1

Dear Dr. Jung,

We’re pleased to inform you that your manuscript has been judged scientifically suitable for publication and will be formally accepted for publication once it meets all outstanding technical requirements.

Kind regards,

Antonio Riveiro Rodríguez, PhD

Academic Editor

PLOS ONE

Reviewers' comments:

Reviewer's Responses to Questions

**Comments to the Author**

1. If the authors have adequately addressed your comments raised in a previous round of review and you feel that this manuscript is now acceptable for publication, you may indicate that here to bypass the “Comments to the Author” section, enter your conflict of interest statement in the “Confidential to Editor” section, and submit your "Accept" recommendation.

Reviewer #1: All comments have been addressed

Reviewer #2: All comments have been addressed

2. Is the manuscript technically sound, and do the data support the conclusions?

Reviewer #1: Yes

Reviewer #2: Yes

3. Has the statistical analysis been performed appropriately and rigorously? 

Reviewer #1: Yes

Reviewer #2: Yes

4. Have the authors made all data underlying the findings in their manuscript fully available?

Reviewer #1: Yes

Reviewer #2: (No Response)

5. Is the manuscript presented in an intelligible fashion and written in standard English?

Reviewer #1: Yes

Reviewer #2: (No Response)

6. Review Comments to the Author

Reviewer #1: I have no comments or suggestions, authors have made all changes suggested earlier and explain them in the cover letter.

Reviewer #2: Many thanks for the revision and incorporating all suggested changes to the manuscript and the quality of the manuscript has been improved.

7. PLOS authors have the option to publish the peer review history of their article (what does this mean?). If published, this will include your full peer review and any attached files.

Reviewer #1: No

Reviewer #2: No

---

## [Editor Report · Acceptance letter]

17 Mar 2021

PONE-D-21-03699R1 

Fracture strength analysis of titanium insert-reinforced zirconia abutments according to the axial height of the titanium insert with an internal connection 

Dear Dr. Jung:

I'm pleased to inform you that your manuscript has been deemed suitable for publication in PLOS ONE. Congratulations! Your manuscript is now with our production department. 

Kind regards, 

on behalf of

Dr. Antonio Riveiro Rodríguez 

Academic Editor

PLOS ONE